# Imitation Learning-Based Energy Management Algorithm: Lille Catholic University Smart Grid Demonstrator Case Study

**Taheni Swibki** [1,*]**, Ines Ben Salem** [1] **, Youssef Kraiem** [2]**, Dhaker Abbes** [2] **and Lilia El Amraoui** [1]

[1] Research Laboratory Smart Electricity and Information and Communications Technologies, SEICT, LR18ES44, National Engineering School of Carthage, University of Carthage Charguia II, Charguia II Tunis-Carthage 2035, Tunisia; ines.bensalem@enicarthage.rnu.tn (I.B.S.); lilia.elamraoui@enicarthage.rnu.tn (L.E.A.)

[2] Arts et Metiers Institute of Technology, University of Lille, Centrale Lille, Junia, ULR 2697-L2EP, F-59000 Lille, France; youssef.kraiem@junia.com (Y.K.); dhaker.abbes@junia.com (D.A.)

* Correspondence: taheni.swibki@enicar.ucar.tn

**Abstract:** This paper proposes a novel energy management approach (imitation-Q-learning) based on imitation learning (IL) and reinforcement learning (RL). The proposed approach reinforces a decision-making agent based on a modified Q-learning algorithm to mimic an expert demonstration to solve a microgrid (MG) energy management problem. Those demonstrations are derived from solving a set of linear programming (LP) problems. Consequently, the imitation-Q-learning algorithm learns by interacting with the MG simulator and imitating the LP demonstrations to make decisions in real time that minimize the MG energy costs without prior knowledge of uncertainties related to photovoltaic (PV) production, load consumption, and electricity prices. A real-scale MG at the Lille Catholic University in France was used as a case study to conduct experiments. The proposed approach was compared to the expert performances, which are the LP algorithm and the conventional Q-learning algorithm in different test scenarios. It was approximately 80 times faster than conventional Q-learning and achieved the same performance as LP. In order to test the robustness of the proposed approach, a PV inverter crush and load shedding were also simulated. Preliminary results show the effectiveness of the proposed method.

**Keywords:** imitation learning; reinforcement learning; linear programming; real-scale microgrid demonstrator; energy management; storage system

## 1. Introduction

The growing trend toward integrating distributed renewable energy resources, such as PV and wind systems, in electrical power grids has a substantial impact on their operation and planning due to their high uncertainty and variability [1,2]. In the last two decades, a new concept, the MG concept, appears as a promising solution to overcome these problems. The term MG is used to describe the concept of intelligent management of energy of a cluster of loads and sources within defined electrical and space boundaries, which can be perceived by the utility grid as a load or generator [3]. An MG provides an alternative solution to reduce the complexity of an electrical system by dividing it into small entities. Moreover, it contributes to decreasing power losses since distributed generators and loads are close to each other, enhancing the power quality and maintaining the balance of the utility grid [4].

One of the most challenging issues in MG operation is the development of an efficient energy management system (EMS) that maintains the MG operation constraints, minimizes energy costs, and ensures the exploitation of distributed energy resources in an optimal way [5]. In this context, the model-based paradigm has shown effective results in solving several energy management problems [6,7]. In this sense, in [8], a day-ahead energy management strategy based on the dynamic programming of a multi-source micro-cogeneration

system was proposed to minimize the cost of fuel consumption. In addition, the authors in this paper point out the advantages of the incorporation of thermal energy storage in the energy management problem. Further, authors in [9] propose a framework to solve an economic dispatch problem under uncertainties in both production and consumption. First, a Fuzzy Monte Carlo simulation was performed to model the system uncertainties. Then, an economic model predictive control algorithm was employed to solve the economic dispatch problem. These algorithms are mainly based on accurate physical models and appropriate modeling of uncertainties (renewable energies, electricity prices, etc.). Consequently, if an accurate model or appropriate estimates of uncertainties mainly relying on domain expertise are unavailable, the energy management algorithm performances will be affected [10].

In recent years, new emerging human-level intelligence algorithms, such as RL and deep reinforcement learning (DRL), have surpassed human expertise in solving several decision-making problems by learning to act in a given environment through experience [11]. Unlike the model-based paradigm, learning-based energy management algorithms can solve the best energy schedule without requiring any prior knowledge of the MG uncertainties, which means that there is no need to use forecasting models to model these uncertainties. This paradigm has been used in several research works related to energy management. In [4], the authors combined diverse machine learning (ML) methods, including RL, transfer learning, and decision trees, to solve an isolated MG economic dispatch problem. The proposed framework was tested using real-world data from the Ecole Polytechnique in France and has shown relevant results in minimizing energy costs without the use of forecasting models. Additionally, batch RL, used in [12], has shown interesting results in optimally controlling an energy storage system (ESS) to enhance the self-consumption of locally produced PV energy and to reduce energy costs under fixed and dynamic electricity prices. In this work, a model-based approach, which is mixed integer linear programming, was employed as a comparison baseline to evaluate the proposed approach. The simulation results have shown a performance gap of 19% between the model-based and the RL-based algorithms. In smart building energy management, RL has been widely used to reduce electricity costs while taking into account users' comfort levels. In [13], a standard RL algorithm, Q-learning, was used to control the energy in a smart home equipped with a rooftop PV system, ESS, and controllable smart home appliances. The simulation results reveal the effectiveness of the proposed approach in minimizing electricity costs under time-of-use pricing. The authors in [14] considered a smart building in a smart grid environment, including a PV system, an ESS, and a vehicle-to-grid (V2G) station. The plug-in of electric vehicles (EVs) adds more uncertainties to the power grid due to their unpredictability and dependency on users' behavior. For this reason, the authors opted for an RL-based approach that has shown satisfying results in solving the best energy scheduling without needing to forecast the electric vehicles charging and discharging behavior during the day. On the other hand, advancements in DRL, which combines the perception power of deep learning models with the ability of RL to learn through interaction with the environment, have shown pioneering results in solving complex MG energy management problems in single and multi-microgrids [15]. In this sense, in [16], the authors introduced a new MG architecture containing innovative components, such as residential price responsive loads and thermostatically controlled load (TCLs) models, to improve MG flexibility. Nevertheless, the inclusion of these components in the energy management problem leads to an increase in its complexity. Consequently, several DRL algorithms were applied to solve the defined MG energy management problem.

Despite the progress made by DRL and RL in MG energy management, it suffers from computation complexity. This is because these algorithms require collecting experience from interacting with the environment over a considerable number of iterations. To overcome this problem, we propose a novel approach based on imitation learning (IL). IL is defined as a sequential decision-making problem where an agent emulates expert demonstrations to learn a behavior as similar as possible to these demonstrations [17]. This

method reduces the complexity of the learning process by leveraging high-quality data samples instead of starting from scratch [18,19]. Imitation learning is divided into two main approaches: behavioral cloning and inverse reinforcement learning [20]. Behavioral cloning IL applied in MG energy management was first introduced in [21]. Authors in this work use a set of demonstrations derived from MILP to learn a policy close to demonstrations. This was achieved by training a deep neural network and reducing the error between the demonstrated and the learned policy. The results showed the effectiveness of the proposed approach, which sped up 17 times the training time in contrast to conventional RL methods. Moreover, the operational cost of the proposed approach is close to the optimal cost. Unlike behavioral cloning, inverse RL tries to learn a reward function based on expert behavior. In [22], inverse RL was employed to accelerate training time by learning a reward function from a rule-based controller demonstration and then using it to train a forward RL algorithm. The proposed method showed promising results in building energy management.

A behavioral cloning-based approach can suffer from a distribution shift, while inverse RL evolves a complex approximation technique to learn a reward function [23]. To overcome these issues, we propose a simple but effective IL-based energy management approach. This approach still uses RL to learn from demonstrations but without requiring heavy approximations to learn a reward function. First, LP is used to build a reference policy, which serves as a demonstration for a modified version of the conventional Q-learning algorithm that is called imitation-Q-learning. The key idea behind the discussed approach is to give the RL agent motivation to match the expert demonstrations. Consequently, while interacting with the environment, the imitation-Q-learning receives a positive constant reward when matching the LP optimal reference in a given state and a negative one otherwise.

This article is organized as follows: A real-scale MG at the Lille Catholic University is presented in Section 2. This demonstrator MG will serve as a case study in the following sections. Then, the energy management problem formulation and the Markov decision process (MDP), which are used as a framework to apply the proposed approach based on IL and a conventional RL approach, are presented in Section 3. The simulation set-up and the main results are discussed in Section 4. Finally, Section 5 concludes this work and opens new perspectives.

## 2. Problem Formulation

### 2.1. Case Study Description

Driven by the social transformation and energy transition context, 108 demonstrators have been deployed across the French territory, the European leader in terms of investment [24]. In 2013, the Haut de France region, inspired by the third revolution concept and introduced by the American economist and sociologist Jeremy Rifkin, launched the project REV3, the third industrial revolution. This project aims to promote a more sustainable region and economy. Motivated by REV3, the Live Tree project includes a plethora of actors collaborating to transform the campus of Lille Catholic University in Vauban city into a living laboratory of social innovation and to reduce the carbon footprint [24,25]. As shown in Figure 1, the Catholic University demonstrator includes two rooftop PV systems, respectively, with 189 kWc and 28 kWc of rated power, a Eaton ESSs (xStorage Building) with a capacity of 250 kWh, four buildings with different load types, six EVs chargers, and a PCC (point of common coupling) to the local power grid of 15 kV through a Fronius inverters [26,27]. This makes the Lille Catholic University smart grid demonstrator a great example of a real-scale MG that serves as a concrete case study in this article.

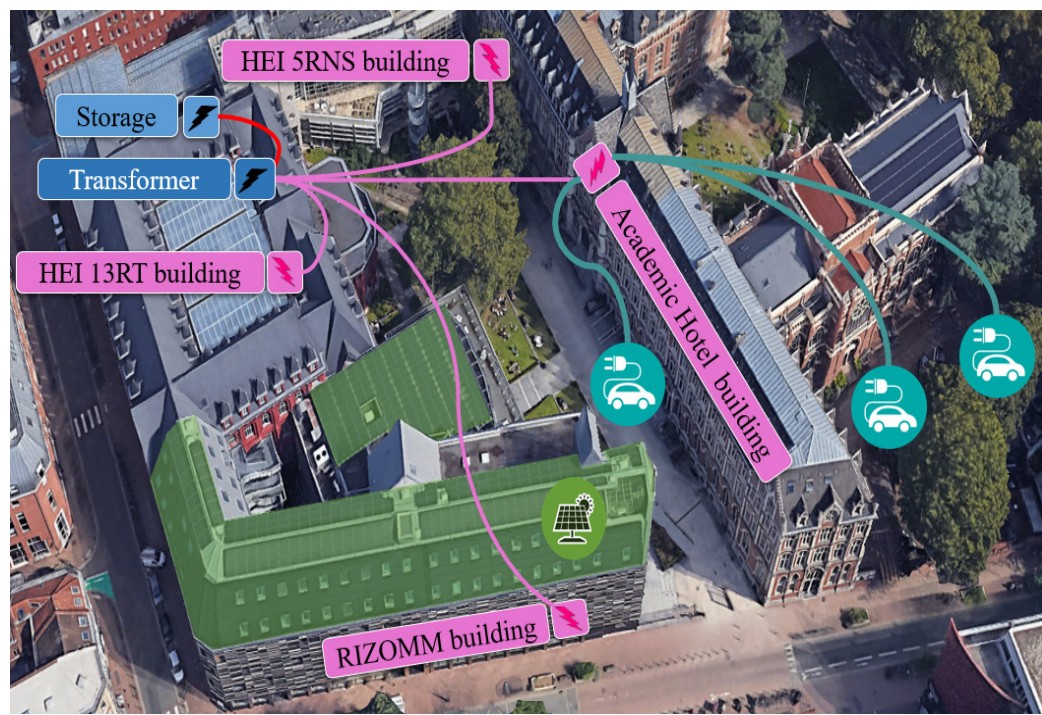

**Figure 1.** Case study: real-scale MG demonstrator at the Lille Catholic University, France.

A schematic representation of the studied MG architecture is given in Figure 2. To model the MG at each time step, only the following power flows will be considered:

- $P_g(t)$: bidirectional power exchanged between the MG and the distribution grid. Therefore, it is assumed that if $P_g(t) > 0$, the MG consumes power from the distribution grid, and if $P_g(t) < 0$, MG injects power into the distribution grid.

- $P_{pv}(t)$: power produced by the PV system.

- $P_B(t)$: considered the charging power of the ESS if $P_B(t) < 0$, and the discharging power if $P_B(t) > 0$.

- $P_{Loads}(t)$: aggregated power demand of the loads.

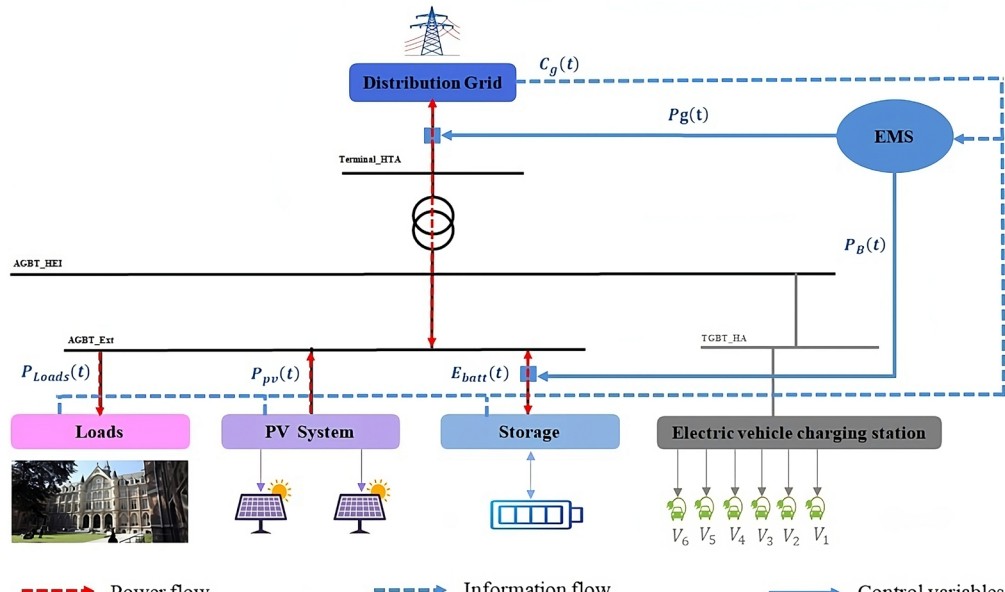

**Figure 2.** MG power flow diagram studied.

### 2.2. Formulation of the Optimization Problem

The objective of the energy management algorithm is to achieve an efficient use of the MG local energy resources to maximize profits. The problem objective function J, which is a function of the grid power exchanged with the MG and the hourly electricity prices, is defined by (1). The PV panels and ESS are exploited to supply the local demand, which can reduce the MG dependency on the power grid and, therefore, reduce the price of purchased electricity. Furthermore, ESSs enable MGs to store electricity during low electricity prices and sell it to the grid at high price periods [28].

$$J = \min \sum_{t=0}^{T} C_g(t) P_g(t) \Delta t \tag{1}$$

where $C_g(t)$ is the electricity costs, $P_g(t)$ is the grid power, T is the horizon of the problem, and $\Delta t$ is the decision time slot.

To solve the predefined energy management problem, several constraints related to the MG power flows should be considered. Consequently, the balance between supply and demand, expressed by (2), must be guaranteed by the energy management algorithm at each time step.

$$P_{pv}(t) + P_g(t) + P_B(t) = P_{Loads}(t) \tag{2}$$

The evolution of the ESS energy level $E_{batt}(t)$ depends on its previous energy level $E_{batt}(t - \Delta t)$ and the supplied/recovered energy amount $P_B(t)\Delta t$ at each scheduling step $\Delta t$:

$$E_{batt}(t) = E_{batt}(t - \Delta t) - P_B(t)\Delta t \tag{3}$$

For safe use of the RSS, the energy level should be maintained within a lower bound and an upper bound, respectively, $E_{batt\_min}$ and $E_{batt\_max}$. This safety constraint is expressed by the following inequality:

$$E_{batt\_min} \leq E_{batt} \leq E_{batt\_max} \tag{4}$$

where these energy limits are defined in Table 1.

**Table 1.** Simulation parameters.

|  | Variable | Value |
|---|---|---|
| Lille Catholic University smart grid demonstrator parameters | Max PV power | $PV_{max} = 217$ kWh |
|  | ESS capacity | $C = 250$ kWh |
|  | Max rate of charge | $P_{charge}^{max} = 40$ kW |
|  | Max rate of discharge | $P_{Discharge}^{max} = 80$ kW |
|  | Max ESS energy level | $E_{batt\_max} = 0.9 \times C$ |
|  | Min ESS energy level | $E_{batt\_min} = 0.1 \times C$ |
| Q-Learning | Number of iterations | N = 100,000 |
|  | Learning rate | $\alpha = 0.1$ |
|  | Discount factor | $\gamma = 0.99$ |
|  | Epsilon | $\varepsilon = 1$ decreasing to 0.01 |
| Imitation-Q-Learning | Number of iterations | N = 100 |
|  | Learning rate | $\alpha = 0.1$ |
|  | Discount factor | $\gamma = 0.99$ |

Furthermore, the recovered/supplied power exchanged with the ESS should be limited by a maximum charge rate $P_{Discharge}^{max}$ and discharge rate $P_{charge}^{max}$ defined by the constructor:

$$P_{Discharge}^{max} \leq P_B(t) \leq P_{charge}^{max} \tag{5}$$

*2.3. MDP—Markov Decision Process Model Formulation*

An MG is considered a continuous, sequential, dynamic, stochastic, and partially observable environment whose evolution depends on the choice of control actions. Thus, its associated energy management problem can be formulated as a sequential decision-making problem under uncertainty, where uncertainties come from the high intermittency of renewable resources, load unpredictability, and electricity price variation. For this purpose, a controlled stochastic process, MDP, which offers a mathematical framework to model such problems, is defined. The MDP is formalized by a tuple $\langle S, A, P, R \rangle$, where S is the MG states, A is an action space, P is a transition model, and R is a reward function.

### 2.3.1. State Space

The state space includes all the accessible information that can provide a valuable basis for the decision-maker to perceive its environment and choose an action to perform. In this study, we assume that the net power demand $P_{net}(t)$, the ESS energy level, and the electricity tariffs are sufficient to model the MG state.

$$S_t = \left\{ P_{net}(t), E_{batt}(t), C_g(t) \right\} \tag{6}$$

where $P_{net}(t)$ models the net power demand of the MG, which is the difference between the PV power and the aggregated residential load power at each decision step.

### 2.3.2. Action Space

The decision variables are the MG power set points that can be affected by the energy management algorithm to satisfy the load consumption at a lower price. The vector of optimization variables, given by (7), contains the references of the ESS power and the power exchanged with the distribution grid.

$$X = \begin{bmatrix} P_B(t) & P_g(t) \end{bmatrix} \tag{7}$$

To ensure a secure operation of the MG, the RL agent controls only the ESS power. Thus, the distribution grid acts as a reserve to compensate for any occurring power imbalance based on the predefined constraint in (2). Therefore, the resulting action space can be defined as follows:

$$A_t = \{\text{charge, Discharge, Idle}\} \tag{8}$$

Once the energy management algorithm decides an action to perform from the action space $A_t$, the next step is to set the exact amount of power to be charged or discharged from the ESS without violating the constraints defined in (3), (4), and (5). Thus, at every time step, the ESS set point can be computed using Equation (9) based on the energy level $E_{batt}(t)$, the energy level constraints $E_{batt_{max}}$ and $E_{batt_{min}}$, and the charging and discharging constraints $P_B^{min}$ and $P_B^{max}$. Subsequently, the distribution grid power reference is computed using the balance equation defined by Equation (2). In this way, all the predefined MG operational constraints will be guaranteed.

$$P_B(t) = \begin{cases} -\min\left( P_{charge}^{max}, \frac{(E_{batt_{max}} - E_{batt}(t))}{\Delta t} \right), & \text{if } a_t = \text{Charge} \\ \min\left( P_{Discharge}^{max}, \frac{(E_{batt}(t) - E_{batt_{min}})}{\Delta t}, & \text{if } a_t = \text{Discharge} \\ 0, & \text{if } a_t = \text{Idl} \end{cases} \tag{9}$$

### 2.3.3. Reward Function

The reward or cost function specifies a scalar feedback signal assigned to a decision-maker for performing a given task. This feedback can be interpreted as negative (punishment) or positive (reward). In this work, the task to be learned by the decision-maker, which is the energy management algorithm, is the minimization of the energy cost used to

operate the MG. Consequently, the reward function, Equation (10), calculates the energy cost of every transition from a state $s_t$ to a next state $s_{t+1}$ by choosing an action $a_t$ from the action space. A positive reward means that the MG sells the excess energy from the PV system and/or from discharging the ESS. Otherwise, a negative reward implies the amount of money to be paid for covering the loads and/or charging the ESS from the distribution grid.

$$R(s_t, a_t, s_{t+1}) = -C(t) \times P_g(t) \times \Delta t \tag{10}$$

## 3. Energy Management Algorithms

### 3.1. Reinforcement Learning

In this section, a model-free RL algorithm, Q learning, is adopted to solve the predefined MDP. As illustrated in Figure 3, the RL agent learns to find a way to act optimally in a given environment by repeatedly testing all actions in all states until determining an optimal policy. The RL agent chooses an action $a_t$ from a predefined action space based on the current state of the environment. Consequently, the environment evolves to a new state and attributes a numerical feedback signal called reward to the agent.

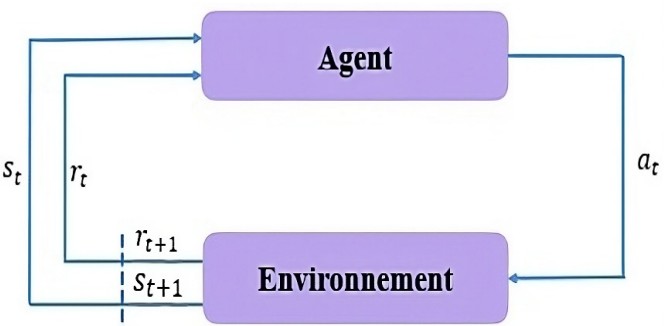

**Figure 3.** Conceptual diagram of reinforcement learning (RL).

The reward signal is a crucial element in the learning process because it defines the quality of the chosen action, indicating whether it is good or bad at a given time step. Rewards in RL problems are analogous to the experiences of pain or pleasure in biological systems [29]. These two feelings are, according to Freud's pain–pleasure principle, mastering humankind decisions and acts by making them seek pleasure and avoid pain. Analogously, the objective of the RL agent is to maximize the expected discounted cumulative reward, given by Equation (11), over time by seeking actions that result in high rewards and avoiding taking low-rewarded actions.

$$R_t = \sum_{k=0}^{\infty} Y^k r_{t+k+1} \tag{11}$$

with $Y \in [0, 1]$ defined as a discount factor, which is used to determine the importance accorded to future rewards. If we set $Y = 0$, the agent, in this case, is "myopic"; thus, it will focus only on maximizing immediate rewards. In contrast, if $Y$ approaches 1, the immediate reward and future long-term rewards will be weighed equally [14,29].

The Q-learning algorithm involves estimating the Bellman equation given by Equation (12). In contrast to rewards that are immediate, the Bellman equation, like the state-action value function, is a recursive equation that predicts the total expected long-term return when starting from a state $s_t$ and taking an action $a_t$ based on a policy $\pi$ [14,29].

$$Q(s, a) = \mathbb{E}_\pi \left[ r_{t+1} + Y \sum_{k=0}^{\infty} Y^k r_{t+k+2} | s_t = s, a_t = a \right] \tag{12}$$

Solving an RL problem means finding an optimal policy that gives the maximum rewards. This optimal policy must satisfy the Bellman optimality equation for state-action values given by (13) [29].

$$Q^*(s, a) = \mathbb{E}_\pi \left[ r_{t+1} + Y \max_{a'} Q^* \left( s_{t+1}, \ a' \right) | s_t = s, a_t = a \right] \tag{13}$$

The learning process of the Q-learning algorithm is summarized in Algorithm 1. At each decision time t in the learning process, the RL agent observes the considered MG state, which is the net power demand, the ESS energy level, and the electricity price. Then, the RL agent takes an action following a given policy and applies it to the MG. This action results in a reward value that will be used to update the Q-value function according to (14).

$$Q(s_t, \ a_t) \leftarrow (1 - \alpha)Q(s_t, \ a_t) + \alpha \left[ R(s_t, \ a_t) + Y \max_{a'} Q \left( s_{t+1}, a' \right) \right] \tag{14}$$

with $\alpha \in [0, 1]$ as the learning rate. If $\alpha$ is fixed as 0, the value function will never be updated, and the agent's new experience will be ignored. Thus, it will never converge to an optimal policy. Otherwise, if $\alpha = 1$, the agent will consider only new state-action Q-values [14].

After finishing the training process, the optimal policy of the RL-based energy management algorithm is obtained from the optimal estimated Q-value function as follows:

$$a_t^* = \pi^*(s_t) = \underset{a}{\mathrm{argmax}} Q^*(s_t, a_t) \tag{15}$$

---

**Algorithm 1:** Q-learning-based energy management algorithm

---

1.  Input parameters: setting the learning rate $\alpha$ and the discount factor Y.
2.  Initialize the action-value function: $Q(s_t, a_t) = 0$
3.  for Episode = 1, M do
4.      Initialize the system state: $S_t = S_0$
5.      for t = 1, T do
6.          Choose an action from the action space using an epsilon-greedy policy
7.          Take action, obtain reward, and observe the next state $S_{t+1}$
8.          Update the action-value function according to (14)
9.          Set $S_t = S_{t+1}$
10.     End for
11.  End for

---

Since learning is based on how the agent interacts with the environment, a policy organizing those interactions is defined. Thus, we define the epsilon-greedy policy given by (16). Using this policy, the agent may choose a random action from the action space with a probability of epsilon or a greedy action that has the highest value with a probability of 1-epsilon. The exploration rate $\varepsilon \in [0, 1]$ is the probability of exploration/exploitation, which is fixed in a way to compromise between exploration and exploitation. When exploring, the agent improves its knowledge of the MG environment, leading to a more accurate state-action value function. In contrast, in exploitation, the agent selects actions that give the highest reward based on its learned action-state value function.

$$a_t = \begin{cases} \text{random} & \text{with probability } \varepsilon \\[2ex] \underset{a}{\mathrm{argmax}} Q(s_t, a_t) & \text{with probability } 1 - \varepsilon \end{cases} \tag{16}$$

### 3.2. Imitation Learning

In the present section, a new learning approach, deployed in Figure 4, is proposed. This approach uses IL to mimic an expert demonstration and RL to learn how to map those optimal actions to the MG states. This algorithm will be called imitation-Q-learning in the rest of the paper. Initially, an expert policy, given by (17), is built by solving the proposed energy management problem modeled by (1)–(5) using LP. Accordingly, the expert will provide the exact theoretical solutions to the optimization problem for several days, given the historical data on production, consumption, and electricity prices.

$$a_t^{\text{Expert}} = \pi^{\text{expert}}(s_t) \tag{17}$$

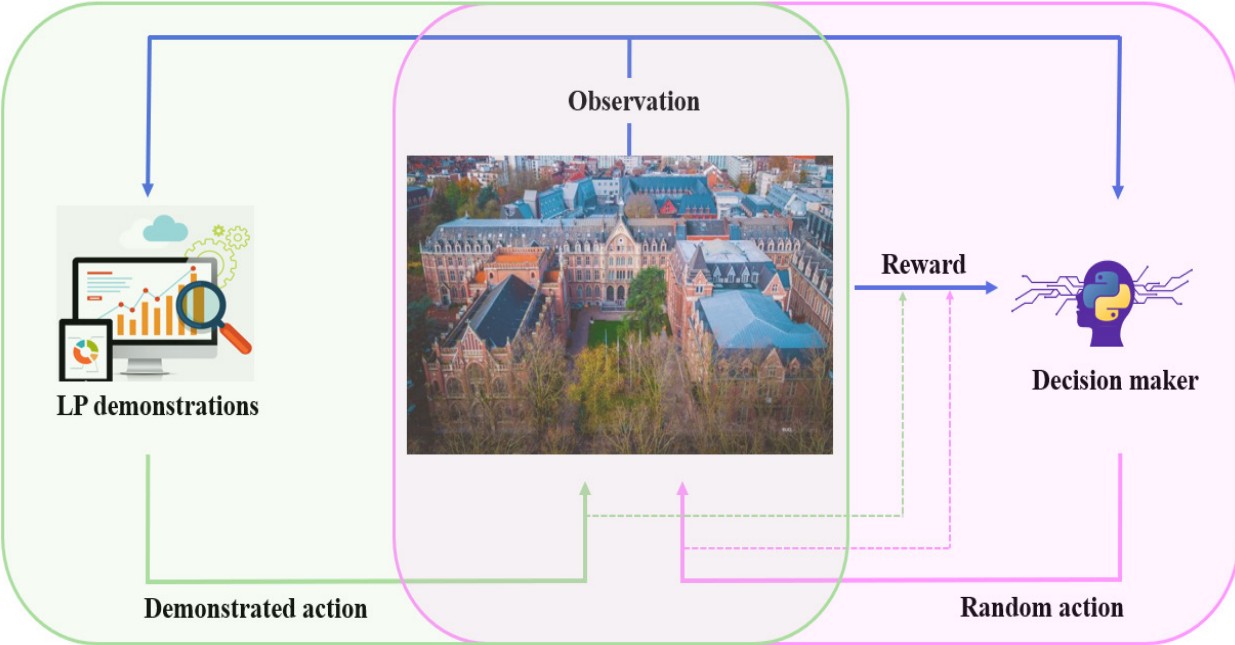

**Figure 4.** Architecture of the proposed energy management approach: imitation-Q-learning.

Similar to RL, the energy management problem is perceived as a sequential decision-making problem under uncertainty. Thus, the same MDP model presented in Section 3 will be used to model the MG dynamics and its interaction with the decision-making agent. The expert policy is used to train the proposed algorithm, which is a slightly modified version of the conventional Q-learning algorithm. As in Q-learning, the proposed algorithm involves searching for an optimal policy that satisfies the optimality equation for state-action values given by (13). The key idea behind this algorithm is to motivate the algorithm to match demonstrations during training using a reward signal given by (18). Consequently, the agent receives a positive reward r = +1 for mimicking the LP optimal action in a given state and a negative reward r = −1 otherwise. To simplify the action space, the imitation-Q-learning algorithm will control only the ESS to simplify the action space and to ensure the handling of the MG operational constraints. Subsequently, only the ESS power set point given by LP will be considered, and the grid power will be computed using the balance Equation (2).

$$r_t = \begin{cases} 1 & \text{if } a_t^{\text{agent}} = a_t^{\text{expert}} \\ -1 & \text{otherwise} \end{cases} \tag{18}$$

Algorithm 2 summarizes the imitation-Q-learning training process. At each decision time t, the imitation-Q-learning receives the MG state and randomly applies an action to the ESS from the predefined action space in (8). Then, if this action matches the expert action, a positive reward is attributed; otherwise, a negative reward is given. The update

rule, given by (19), is employed to update the imitation-Q-learning action-value function. In the proposed approach, it is important to ensure the synchronization between the expert demonstrations and the imitation-Q-learning and avoid ending up in two different states. Thus, the expert action will be applied to control the MG dynamics instead of the imitation-Q-learning actions. At the end of the training, the imitation-Q-learning algorithm will be able to control the MG using a derived policy from the learned optimal action-value function.

$$a_t^{\text{Imitation}-Q-\text{learning}^*} = \underset{a}{\operatorname{argmax}} Q^*(s_t) \tag{19}$$

---

**Algorithm 2:** Imitation-Q-learning-based energy management algorithm

---

1.      Load expert policy
2.     Imitation-Q-learning input parameters: setting the learning

   rate $\alpha$ and the discount factor Y.

3.      Initialize the action-value function: $Q(s_t, a_t) = 0$
4.     for Episode = 1, M do
5.          Initialize the system state: $S_t = S_0$
6.          for t = 1, T do
7.              Derive the expert action: $a_t^{\text{Expert}} = \pi^{\text{expert}}(s_t)$
8.              Choose a random action from the agent action space
9.              Compare the agent action to expert action, obtain reward, and observe the next

   state

$$S'_{t+1}.$$

10.              Update the action-value function according to (14)
11.              Move to a next state $S_{t+1}$ based on the expert action
12.     Set      $S_t = S_{t+1}$
13.              End for
14.     End for

---

## 4. Results and Discussion

### 4.1. Simulation Data and Set-Up Parameters

In this section, the MG demonstrator at the Lille Catholic University is used as a case study to conduct the experiments. Since we are working on learning-based approaches, historical data are needed to train the studied algorithms. Figure 5 reveals the PV power and consumption profile during the COVID-19 pandemic every ten minutes from 1 January 2020 to 31 December 2020. This historical data shows clearly that several unpredictable factors have a major impact on the power system. As an example, restrictions related to the COVID-19 pandemic have dramatically changed the consumption profile on the Catholic campus. Furthermore, several technical faults can occur and change production and consumption behaviors. As a concrete example of these faults, the PV inverter crashed in the summer of 2020 while trying to inject power into the distribution grid.

The catholic university follows a time-of-use (ToU) energy pricing policy with seasonal variation [30]. As inputs in the training process, only data from January 2020 is considered; accordingly, the winter energy pricing policy, Equation (20), is used. The peak load period is from 07:00 a.m. to 22:00 p.m., and the off-peak load period is from 23:00 p.m. to 06:00 a.m.

$$C_g(t) = \begin{cases} 18.04 \ (c€/kWh) & \text{if peak loads} \\ 12.26 \ (c€/kWh) & \text{if off} - \text{peak loads} \end{cases} \tag{20}$$

To train the Q-learning algorithm, a simulation model of the studied MG demonstrator was established based on the predefined MDP using the Python programming language on Google Collaboratory (Python 3.10). As mentioned above, data from January 2020 was

used within the pricing policy described by (20) to simulate the microgrid states. During the training process described by Algorithm 1, the Q-learning agent interacts with the simulation model and explores the MG states over several iterations until converging to an optimal policy.

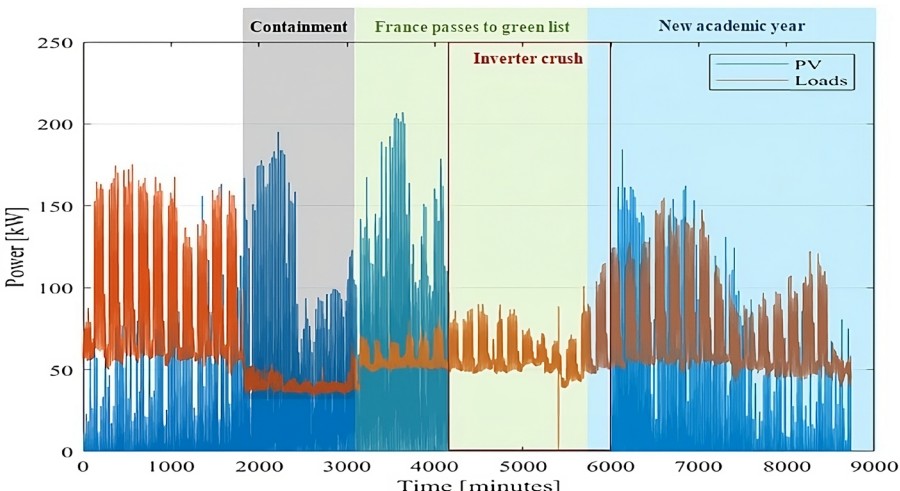

**Figure 5.** Historical data of the MG demonstrator at the Lille Catholic University, France.

To create the expert demonstrations, the Matlab 2023 optimization toolbox and the same historical data were used. Thus, the MG energy management problem was solved day by day using LP. To train the imitation-Q-learning algorithm, the same simulation model used to train the Q-learning algorithm was employed. All the simulation parameters and hyperparameters are summarized in Table 1.

### 4.2. Simulation Results and Discussion

In order to look closely at the proposed energy management approach in realistic situations, two different days are derived from the MG demonstrator's historical data and used as test scenarios. The first is Monday, 17 February 2020, and the second is Thursday, 20 February 2020. According to the weather history in Lille [31], in the first test scenario, the sky was almost partly cloudy. As shown in Figure 6, the PV production exceeds the load power only for three hours in the middle of the day. In the second test scenario, the sky was mostly cloudy. Consequently, the load power was always higher than the PV power during the whole day.

LP is used here as a comparison baseline. The drawback of this method is that it is based on perfect forecasts of the MG variables. Thus, we need to reoptimize the control set points in real time to deal with errors between measured and forecasted values. This article assumes that it has perfect forecasts of the MG variables, such as the PV power, the load demand, and the electricity prices of the next 24 h. The results of applying LP in the two scenarios are presented in Figure 7. At the beginning of the day, the ESS energy level was at its lowest limit. By 6 a.m., which is the end of the off-peak load, the ESS was fully charged from the distribution grid. When electricity prices increased during the peak load period, it was discharged to supply the local demand. The distribution grid was used as a reserve to avoid power imbalance in the MG. Subsequently, the MG consumed power from the grid at low PV production and when the ESS was unavailable or could not cover all the load demand. In contrast, it sold the excess PV production to the power grid. Furthermore, if the load power was below the ESS discharge rate, the excess power was sold to the power grid.

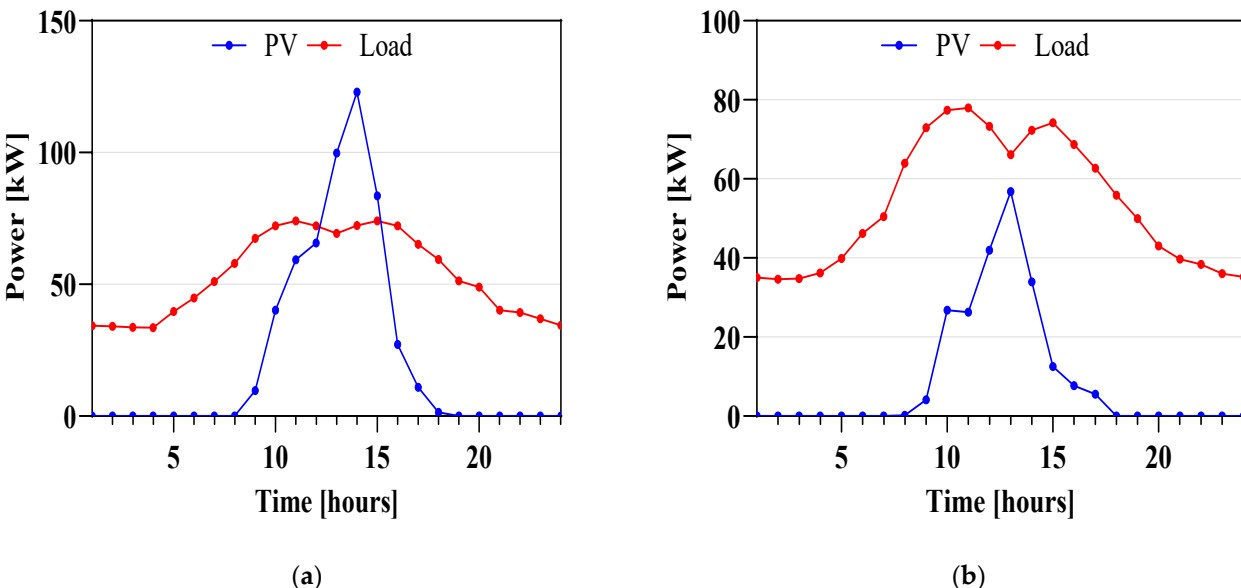

**Figure 6.** Test scenarios: (**a**) scenario 1—Monday 17 February 2020; (**b**) scenario 2—Thursday 20 February 2020.

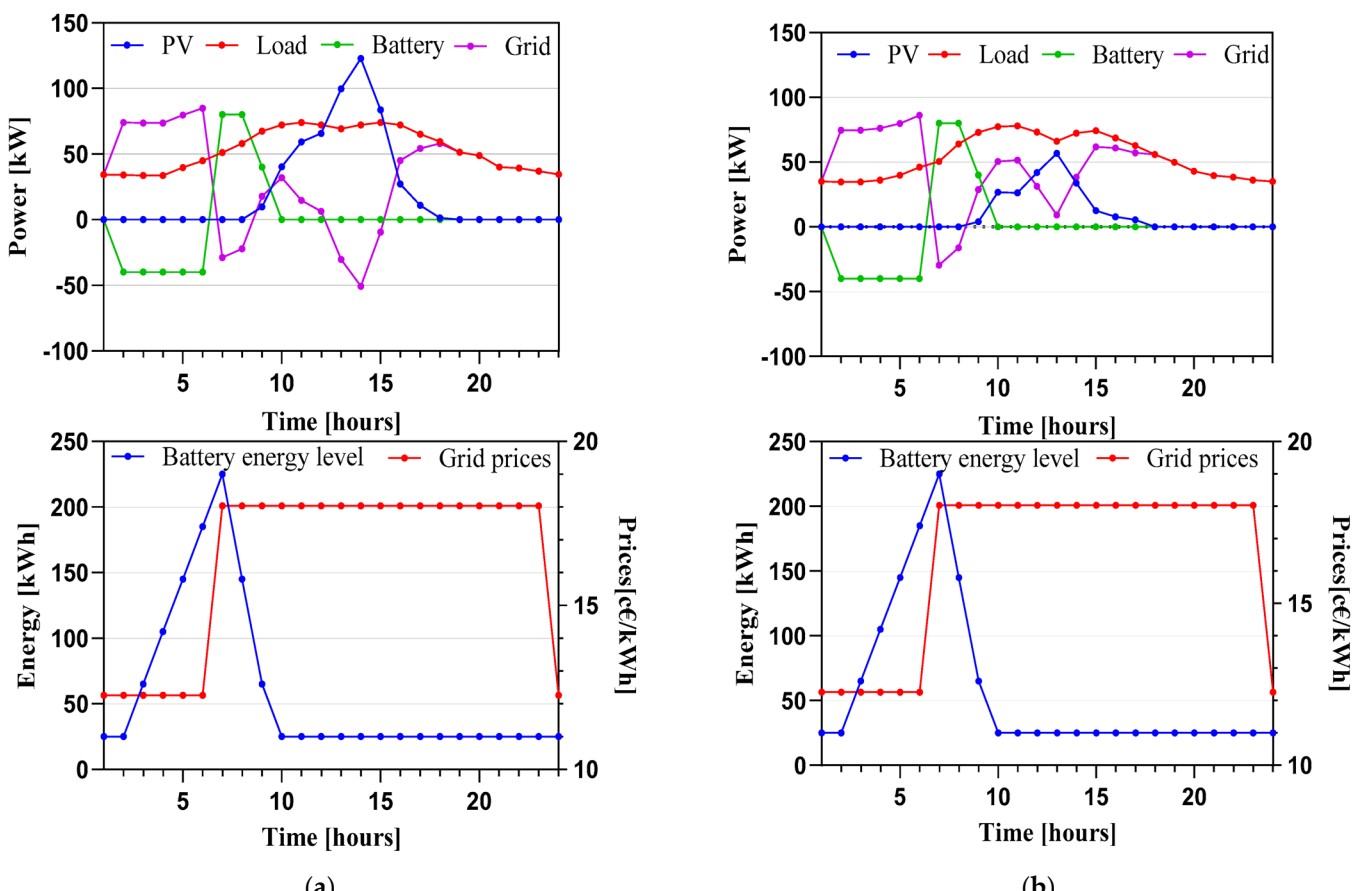

**Figure 7.** LP results—MG powers, ESS energy level, and electricity prices: (**a**) test scenario 1 and (**b**) test scenario 2.

Figure 8 reveals the learning curve of the Q-learning algorithm. During the training process, the total rewards of each episode were computed. To show the reward dynamics more clearly, the average episodic reward was calculated over the last 200 episodes. In this

work, an annealing epsilon was used, which means that a high exploration rate, $\varepsilon = 1$, was used at the beginning and decreased progressively to 0.01. In this way, the agent moved from an explorative behavior to an exploitative one, which helped in achieving a faster convergence. It can be observed that the average reward curve converged to its highest value after approximately 50,000 episodes when epsilon reached its lowest value $\varepsilon = 0.01$.

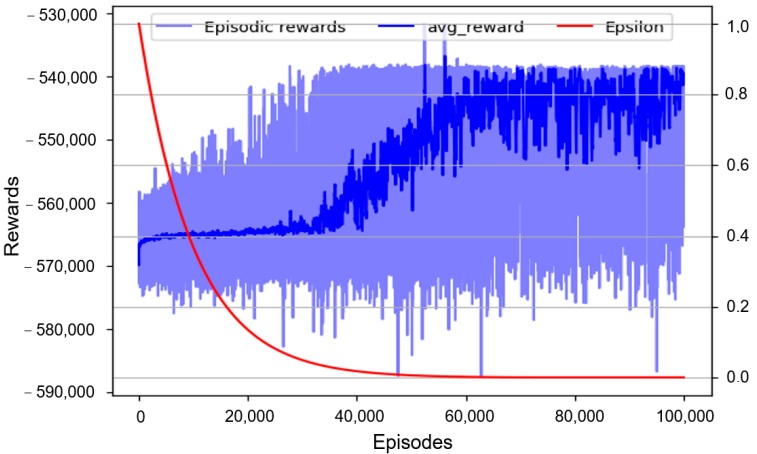

**Figure 8.** Learning curves of the Q-learning algorithm.

Figure 9 depicts the results of applying the Q-learning algorithm to control the demonstrator MG for 24 h. This algorithm could be applied in real time without the need for forecasting models. At each decision time, the Q-learning receives measures of the PV power, the load demand, and the price of energy. Based on this information, a control action is applied to the ESS derived from the learned control policy. Similar to LP results, the ESS was fully charged at low electricity prices at the beginning of the day and discharged to satisfy the loads when electricity prices increased. Furthermore, it is clear from simulations that there is no violation of the ESS security constraints defined by (3-4-5). The Q-learning algorithm is designed for problems with discrete action and state spaces. This means that it cannot deal with continuous and complex problems such as MG energy management. Subsequently, the action space and the state space need to be discretized. Furthermore, it suffers from the curse of dimensionality. Thus, if the dimensions of the state space and the action space increase, the complexity of the problem increases, which leads to slow convergence or even divergence. Further, this algorithm is sensitive to the learning hyperparameters. These hyperparameters are empirically set through trial and error, and it is crucial to show the best hyperparameters depending on the studied problem. As an example, in our case, we used a high discount factor ($Y = 0.99$) to ensure that the algorithm takes into account long-term rewards. Accordingly, it sacrifices immediate rewards when charging the ESS at the beginning of the day to gain long-term higher rewards when discharging it at high electricity prices in the peak load period. In contrast, lower discount factors could make the learning process inefficient. This leads to instability in the algorithm decisions, which could explain the fluctuations in the ESS actions, which in turn has an impact on the grid power. The training process is controlled through empirically defined hyperparameters that have a major impact on the algorithm performance.

As shown in Table 2, the first notable result is that the proposed algorithm, imitation-Q-learning, needed only 100 iterations, which took less than one minute, to learn how to successfully imitate the LP algorithm. The proposed algorithm combines the advantages of LP and Q-learning. It was able to control the MG in real time without the need for prior knowledge about the demonstrator state. On the other hand, it has the same performance as LP, which gives the optimal theoretical solutions to the energy management problem. This makes the proposed approach simpler and more effective than conventional Q-learning. The results in Figure 10, which are identical to LP results, confirm the effectiveness of the proposed imitation-Q-learning algorithm.

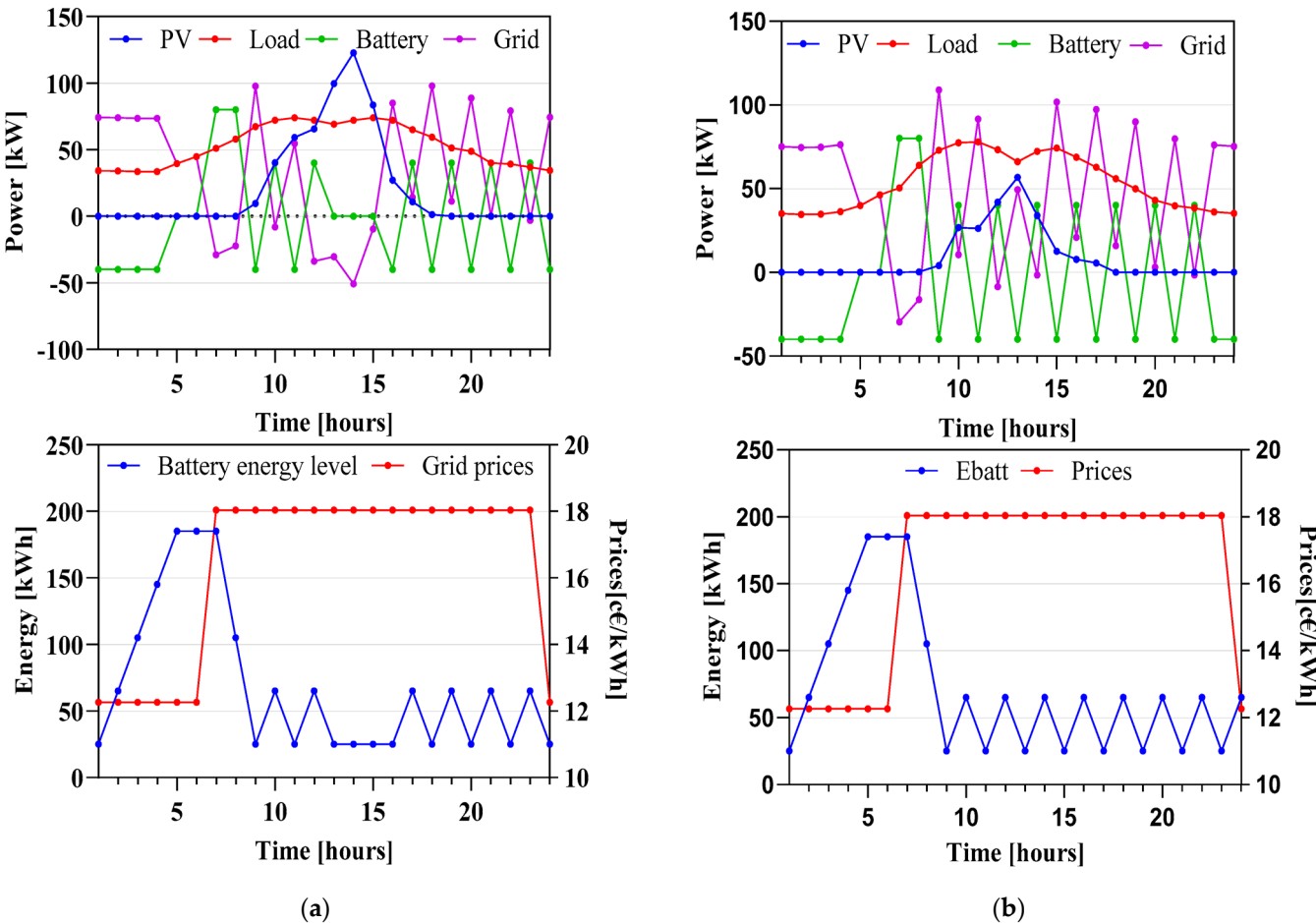

**Figure 9.** Q−learning results—MG powers, ESS energy level, and electricity prices: (**a**) test scenario 1 and (**b**) test scenario 2.

**Table 2.** Simulation time.

| Algorithm | N-Iterations | Simulation Time |
|---|---|---|
| Q-learning | 100,000 | 83.96 min |
| Imitation-Q-Learning | 100 | 58 s |
| LP | - | 1.6 s |

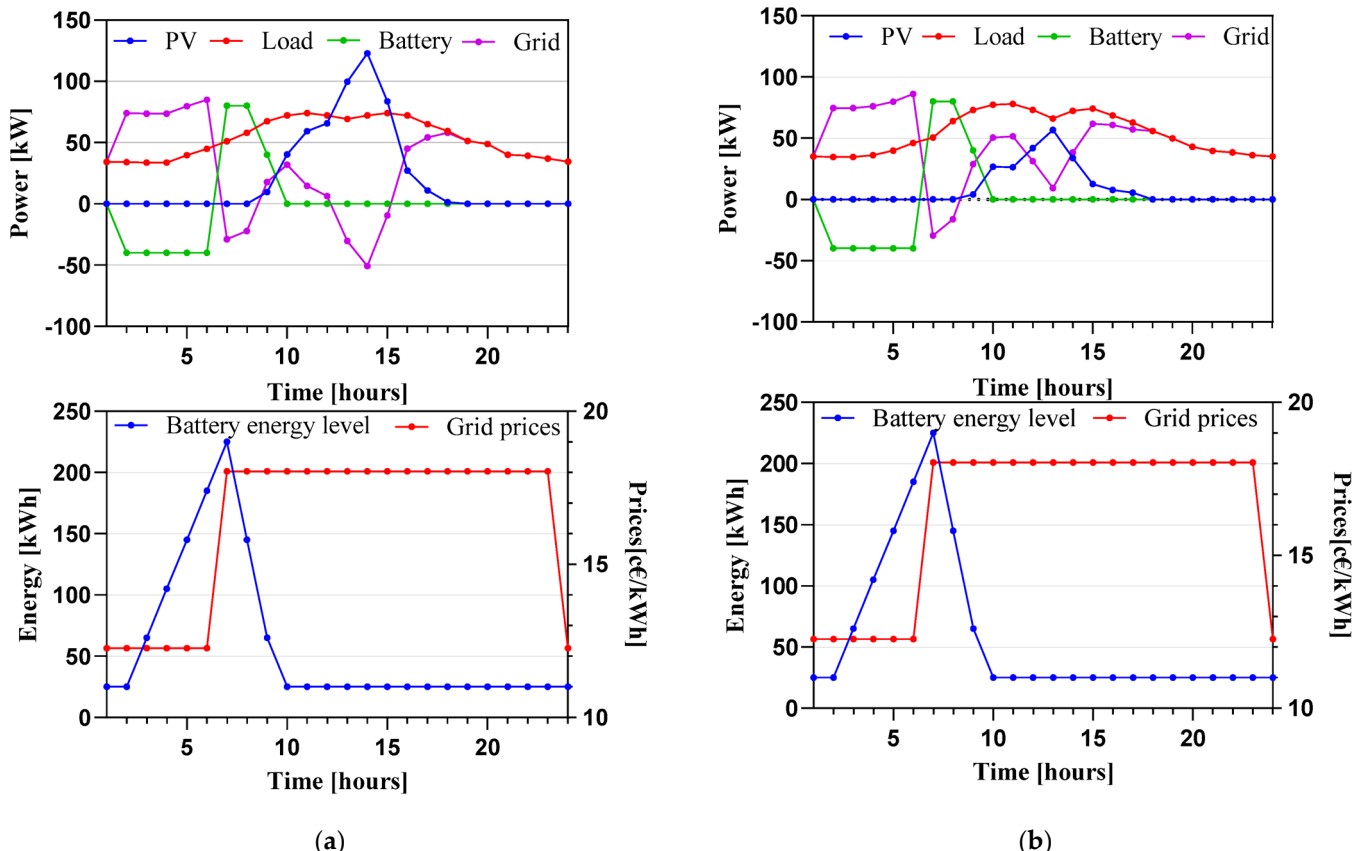

**Figure 10.** Imitation−Q−learning results—MG powers, ESS energy level, and electricity prices: (**a**) test scenario 1 and (**b**) test scenario 2.

To prove the robustness of the proposed approach against uncertain factors, two other simulation scenarios, scenario 3 and scenario 4, were adopted. Scenario 3 reproduces a PV inverter crush similar to the accident that occurred in Lille Catholic Campus in the summer of 2020 when they tried to inject power into the power grid. Consequently, there was no PV power during the whole day. In scenario 4, a load shedding was programmed for 3 h during the peak load period. According to Figure 11, the most notable result is that the proposed algorithm successfully managed the two accruing events in scenarios 3 and scenario 4. Furthermore, the MG balance was guaranteed with no ESS constraint violation.

Figure 12 depicts the cost of the daily energy exchanged with the power grid during scenario 1 and scenario 2. According to the following bar chart, the lowest costs of energy in both scenarios resulted from using LP, which are, respectively, 165.955 in scenario 1 and 110.444 in scenario 2. To compare the three energy management approaches, a performance gap metric, given by (20) [14], is employed. Since LP gives the theoretical solution to the energy management problem, it was used as an optimal reference to compute the cost gap. The cost of energy resulting from using the imitation-Q-learning algorithm is similar to the energy cost when using LP. Consequently, the gap between LP and imitation-Q-learning results is equal to 0% in both scenarios. In contrast, the cost of using Q-learning is 6.5% higher than LP and imitation-Q-learning in scenario 1 and 8.6% in scenario 2.

$$G = \frac{\text{Cost(Algorithm)} - \text{Cost(optimal)}}{\text{Cost(optimal)}} * 100 \tag{21}$$

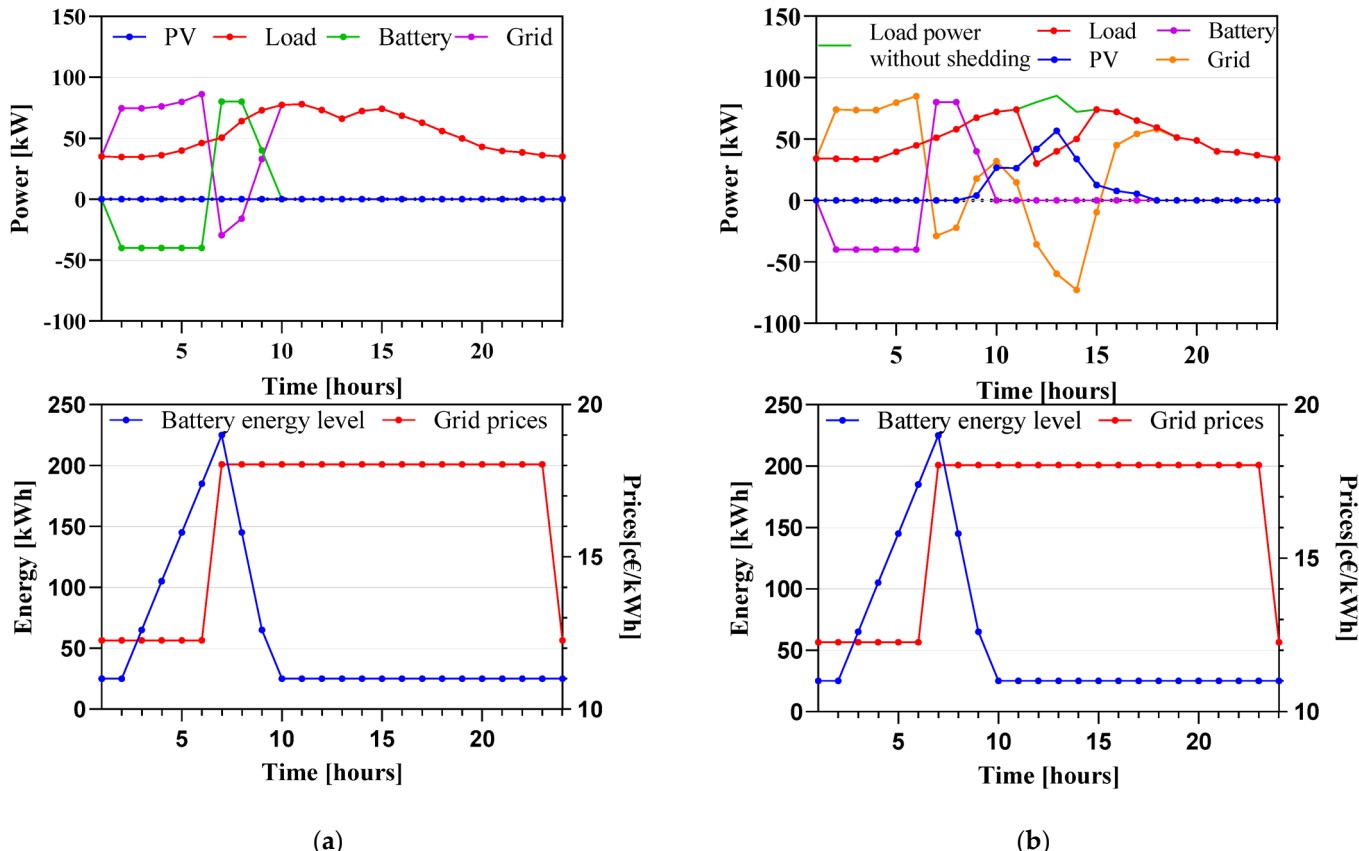

**Figure 11.** Imitation−Q−Learning results—MG powers, ESS energy level, and electricity prices: (**a**) test scenario 3 and (**b**) test scenario 4.

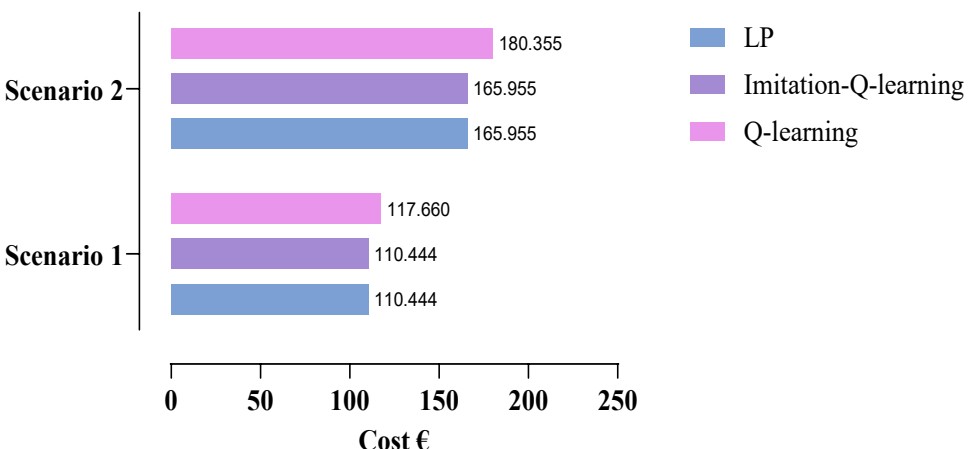

**Figure 12.** Electricity cost per day.

## 5. Conclusions

This article presents a novel IL-based energy management approach called imitation-Q-learning for MG energy management. This approach combines the advantages of LP, IL, and RL to solve the energy management problem of a real-scale MG demonstrator at the Lille Catholic University, France. Real data of production, consumption, and electricity prices for one month are used to solve a set of LP problems, which serve as demonstrations to train the imitation-Q-learning. The resulting policy is employed to control the studied MG without requiring prior information about the MG demonstrator state. Moreover, the experiments showed promising results compared to LP and conventional Q-learning

under different simulation scenarios. The proposed solution speeds up the training process nearly 80 times compared to Q-learning. Moreover, the daily operation cost when using imitation-Q-learning is similar to the cost of LP in all test scenarios.

In this work, it was assumed that all loads are uncontrollable, which is not the case in real life, especially with the renovation of the RIZOMM building, making it a real-scale smart connected building. From this perspective, controllable loads and the electric vehicle charging station of the Catholic campus will be considered. This will add more uncertainties and, thus, new challenges to the proposed energy management algorithm. Furthermore, the effect of the studied energy management approach solution on the demonstrator MG physical properties, such as the voltage constraints, will be added to the MG operational constraints.

**Author Contributions:** Conceptualization, T.S., I.B.S., Y.K., D.A. and L.E.A.; methodology, T.S., I.B.S., Y.K., D.A. and L.E.A.; software, T.S.; validation, T.S., I.B.S., Y.K., D.A. and L.E.A.; formal analysis, T.S., I.B.S., Y.K., D.A. and L.E.A.; investigation, T.S.; resources, Y.K. and D.A.; data curation, T.S.; writing—original draft preparation, T.S.; writing—review and editing, T.S., I.B.S., Y.K., D.A. and L.E.A.; visualization, T.S.; supervision, I.B.S. and L.E.A. All authors have read and agreed to the published version of the manuscript.

**Funding:** This research received no external funding.

**Data Availability Statement:** Data available on request due to privacy.

**Conflicts of Interest:** The authors declare no conflict of interest.

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
