# Peer review of "Imitation Learning-Based Energy Management Algorithm: Lille Catholic University Smart Grid Demonstrator Case Study"

_electronics, doi:10.3390/electronics12245048_

Round 1

Reviewer 1 Report

Comments and Suggestions for Authors

1.       In 3.2.3, the reward function calculates the energy cost. However, the system involves the charging and discharging of the battery. The dynamic of the state of charge (SOC) needs to be considered.

2.       The authors need to provide more details of the training process of Q-learning. Additionally, based on the results in Figure 7, although the exploration rate has been reduced to a very low level, the reward fluctuations are still significant. The results can not sufficiently demonstrate the effectiveness of training progress.

3.       Some DRL methods and their advantages are mentioned in Introduction. However, this paper focuses on the study of the Q-learning method. In fact, Q-learning, as a discrete RL algorithm, may not exhibit as effective performance as other DRL algorithms, such as DDQN and DDPG.

4.       The titles of sections 5.1 and 5.2 are identical.

5.       Some latest research articles and review paper are suggested to be included in the Introduction . See “Machine Learning and Data-Driven Techniques for the Control of SmartPower Generation Systems: An Uncertainty Handling Perspective” and “Sustainable Residential Micro-Cogeneration System Based on a Fuel Cell Using Dynamic Programming-Based Economic Day-Ahead Scheduling”.

6.       The second chapter contains relatively less content; it is advisable to supplement additional descriptions and introductions of system components.

7.       The paper writing is not sufficiently standardized, with some grammar and spelling errors. For instance, the definition of “ESS” is not given.

8.       The quality of figures and their corresponding titles needs improvement.

9.       The formulation of Bellman equation may be incorrect and needs to be corrected.

10.   Section four provides a comprehensive introduction of Q-learning, but there is insufficient elaboration on the integration of Q-learning with linear programming. It is advisable to augment additional details regarding the incorporation of IQ-learning.

11.   The fifth section's content is overly intricate and lacks explanations for scenario 1 and scenario 2.

Comments on the Quality of English Language

major

Reviewer 2 Report

Comments and Suggestions for Authors

This manuscript introduces a novel learning approach that integrates imitation learning with reinforcement learning. The proposed method has been successfully applied to a case study, demonstrating its effectiveness. While the manuscript is well-organized, the quality of writing could be enhanced. Additionally, using higher resolution figures would enhance clarity. Besides, a few minor concerns to be addressed before the acceptance.

1. Line 353. Incorporating a more detailed discussion about stability and efficiency would enhance this portion.

2. In Table 2. The number of iterations is only about 1/1000 (= 100000/100), while the simulation time is about 1/87 (=83.96 min/58 sec). Why they differs such a lot? Are there other factors that have a greater impact on simulation time than the learning aspect?

Comments on the Quality of English Language

The writing can be improved.

Reviewer 3 Report

Comments and Suggestions for Authors

Review#1: This paper proposes a novel energy management approach based on Imitation Learning and Reinforcement Learning, and a real scale Microgrid, Lille Catholic University smart grid demonstrator, was used as a case study to conduct experiments. The following are my comments:

1. The structure of this paper should be modified, such as section 2 and section 5 should be merged.

2. Why are voltage constraints not considered? It is very important for the safe operation of microgrids. In addition, I suggest the authors establish a more detailed model.

3. In section 3.2, the authors should explain how the constraints are handled.

4. Almost all figures are not clear, and the font should be revised to Times New Roman. The authors should check the whole manuscript.

5. In the figure 8, why is the power change significant within 24 hours?

6. In Table 2, the authors should add the time consumed by the linear programming method.

7: More related DRL research should be compared, such as  (not mine): 1:Deep Reinforcement Learning for Microgrid Operation Optimization: A Review 2: Multi-Stage Real-Time Operation of a Multi-Energy Microgrid With Electrical and Thermal Energy Storage Assets: A Data-Driven MPC-ADP Approach 

Comments on the Quality of English Language

can be improved

Round 2

Reviewer 1 Report

Comments and Suggestions for Authors

The revised paper is acceptable for publication. 

Comments on the Quality of English Language

The revised paper is acceptable for publication.